# Smelling TNT: Trends of the Terminal Nerve

**DOI:** 10.3390/ijms25073920

**Published:** 2024-03-31

**Authors:** Wael Abu Ruqa, Fiorenza Pennacchia, Eqrem Rusi, Federica Zoccali, Giuseppe Bruno, Giuseppina Talarico, Christian Barbato, Antonio Minni

**Affiliations:** 1Department of Sense Organs, Sapienza University of Rome, Policlinico Umberto I, 00161 Roma, Italy; aburuqa.1796600@studenti.uniroma1.it (W.A.R.); pennacchia.1748833@studenti.uniroma1.it (F.P.); federica.zoccali@uniroma1.it (F.Z.); 2Department of Human Neuroscience, Sapienza University of Rome, 00185 Rome, Italy; eqrem.rusi@uniroma1.it (E.R.); giuseppe.bruno@uniroma1.it (G.B.); giuseppina.talarico@uniroma1.it (G.T.); 3Institute of Biochemistry and Cell Biology (IBBC-CNR), Sapienza University Rome, Policlinico Umberto I, 00161 Roma, Italy; 4Division of Otolaryngology-Head and Neck Surgery, ASL Rieti-Sapienza University, Ospedale San Camillo de Lellis, 02100 Rieti, Italy

**Keywords:** terminal nerve, olfactory dysfunctions, Kallmann syndrome, GnRH, SARS-CoV-2

## Abstract

There is very little knowledge regarding the terminal nerve, from its implications in the involvement and pathogenesis of certain conditions, to its embryological origin. With this review, we try to summarize the most important evidence on the terminal nerve, aiming to clarify its anatomy and the various functions attributed to it, to better interpret its potential involvement in pathological processes. Recent studies have also suggested its potential role in the control of human reproductive functions and behaviors. It has been hypothesized that it plays a role in the unconscious perception of specific odors that influence autonomic and reproductive hormonal systems through the hypothalamic–pituitary–gonadal axis. We used the PubMed database and found different articles which were then selected independently by three authors. We found 166 articles, of which, after careful selection, only 21 were analyzed. The terminal nerve was always thought to be unimportant in our body. It was well studied in different types of animals, but few studies have been completed in humans. For this reason, its function remains unknown. Studies suggest a possible implication in olfaction due to the anatomical proximity with the olfactive nerve. Others suggest a more important role in reproduction and sexual behaviors. New emerging information suggests a possible role in Kallmann syndrome and COVID-19.

## 1. Introduction

The terminal nerve was first discovered in 1878 by Frisch in elasmobranchs, a subclass of Chondrichthyes, cartilaginous fish. Years later, it was also discovered in humans. More precisely, Johnston described it in human embryos in 1913 and in human adults the following year. Simultaneous with discoveries about its anatomical course and embryological studies, there had been a succession of names attributed to this nerve. Initially, it was named the following: nerve of Pinkus, tractus olfacto-commissuralis, new nerve, terminal nerve, nerve nulla (i.e., nothing, zero), and cranial nerve 13. Then, it was renamed “nervus terminalis” since it entered the region of the lamina terminalis, which is the currently accepted nomenclature [1]. The Latin name nervus terminalis, which has now been replaced by the names terminal nerve and terminalis nerve (TN), refers to a rudimentary structure found in human and higher mammals, which can be found in fetal stages. Still, there is a theoretical interest in its function [2,3].

It is a highly preserved and versatile nerve, located just above the olfactory bulbs in humans and different vertebrate species. In most instances, its fibers extend from the front part of the brain to the olfactory and nasal epithelia. As the name suggests, this nerve penetrates the lamina terminalis of the forebrain, the slender layer of gray matter above the optic chiasm, forming the medial section of the anterior wall of the third ventricle [4] (Figure 1). In its initial segment, it runs within the nasal cavity, being one of the five systems that innervate the nasal cavity (the terminal system, the vomeronasal system, the olfactory system, the septal organ, and the trigeminal system). The fourth system originates from the original olfactory placode [4]. Within the nasal cavity, its fibers intermingle with those of the vomeronasal and olfactory nerves, respectively. However, the terminal nerve fibers travel medial to the nerve fibers of the 1st cranial nerve and into the anterosuperior portion of the nasal cavity. The fibers of the TN branch from the ganglion cells, towards the olfactory bulb and, intracranially, its fibers run independently. Studies conducted on embryos using high-resolution imaging have shown that medially the TN fibers are near the vomeronasal (VN) fibers and laterally, the olfactory fibers reach the olfactory bulb [4]. The terminalis nerve and the vomeronasal nerves penetrate through the cribriform plate, where the plexiform fibers of the terminalis nerve can be distinguished along the ganglion cell bodies that form the terminal ganglion. The plexus of the terminalis nerve is formed by numerous smaller strands that branch and anastomose with each other [2]. Ventrally on the surface of the brain and laterally to the olfactory bulbs, the terminalis nerve and the vomeronasal nerves resume their course along the fibers of cranial nerve I. The terminalis nerve plexus runs parallel to the olfactory tracts in the vicinity of the septal region, near the bifurcation of the anterior and middle cerebral arteries. Its fibers are non-myelinated and run alongside the dura mater, passing through the subarachnoid space and ultimately connecting to the pia mater in the gyri of the frontal lobes. It enters the brain through the lamina terminalis, implying penetration into the prosencephalon [1] (Figure 1). Roussel et al. have confirmed this in their work on the landmarks used during endonasal skull base surgery [5]. Many reports suggest that the terminal nerve projects various neuroanatomical structures, such as the medial pre-commissural septum, including the medial septal nucleus. It also sends fibers to the nasal mucosa and ventral rostral brain structures, mainly in olfactory and limbic areas (i.e., amygdala, hypothalamic nuclei) [6]. We have learned very little about this nerve since its discovery to the present, so it is not mentioned in most anatomy textbooks. Most studies on the TN have been conducted in animals, and a broader view of the TN in humans is important. In this brief review, we try to summarize what we know about this nerve, its anatomy, the hypotheses regarding its functions, and its possible pathological implications in human diseases.

## 2. Materials and Methods

For this research, the “PubMed” database was used. The terms used were as follows: (nervus terminalis OR Terminal Nerve OR cranial nerve 0 OR cranial nerve XIII). This search resulted in 166 possible articles. All non-English language articles and articles for which the full text was not available were excluded. We also had to exclude all the non-free studies that could not be accessed by Sapienza’s institutional credentials. Of the remaining articles, three independent reviewers (F.P., W.A.R., and E.R.) checked them by titles and abstracts, selecting those relevant to the review topic. In this phase, most of the excluded articles were performed on animals without important correlations/implications for humans. We did not use a filter for human studies it our PubMed search because we did not want to risk excluding studies performed on animals with possible repercussions for humans. In the end, 145 studies were excluded, while the remaining 21 were analyzed and are discussed in this review. (Figure 2) (Table 1).

## 3. Results

### 3.1. Embryology

The specific embryological origin of the TN remains partly enigmatic. While some authors have reported that the nerve’s origin is from the olfactory placode, where olfactory cells also originate, others indicate that it arises from the neural crest. The most widely accepted hypothesis is that, like other cranial nerves, its embryological origins lie in synergistic interactions during development between the neural crest and sensory placodes [1]. The terminalis nerve forms at the limit of migrating neural crest cells with the olfactory and adenohypophyseal placodes [4]. Furthermore, the neural crest may contribute to the subset of GnRH-secreting neurons [6].

### 3.2. Neurophysiology and Functional Aspects

Although there are different ideas about the functions of this nerve, in this paragraph we will delve into its potential functions (Figure 3). It is thought that the nerve does not play a role in the olfactive function but in the reproductive function. It secretes luteinizing hormone-releasing hormone (LHRH), which has been associated with reproductive behavior. It is speculated that the nerve’s function in humans is to directly detect or more probably to modulate the activity of the olfactory epithelium, making pheromones more detectable. Pheromones are species-specific odors involved with sexual identification and arousal and therefore are important for mate selection. For example, the woman’s sense of smell is most acute when she is ovulating. Odors are perceived differently throughout the menstrual cycle, making the same odors more pleasant and causing sexual arousal [7]. On the other hand, male odors have been shown to increase male sociosexual behavior. Also, a practical effect can be the phenomenon of women living together who tend to synchronize their menstrual cycles [7]. The TN may trigger hormonal responses, independently or together with other circuits, such as the kisspeptin neural network (mainly localized in the preoptic and infundibular regions of the hypothalamus) [7]. Also, the hormone GnRH/LHRH appears to serve as a coordinating system for the multitude of events occurring during reproduction, including changes in olfactory sensitivity to pheromones [4,7]. Luteinizing hormone-releasing hormone (LHRH) regulates the secretion of both the luteinizing hormone (LH) and follicle-stimulating hormone (FSH) from pituitary gonadotropic cells. The TN shows a similar distribution of LHRH in both juvenile and adult animals. However, most of the activity of LHRH is greater in the adult brain [8].

### 3.3. Neuronal Immunochemical Studies of the Nervus Terminalis

Many studies on the fetal nervous system of animals have shown the presence of gonadotropin-releasing cells. The presence of gonadotropic cells on nerve fibers was analyzed with GnRH immunoreactive molecular techniques. In the hypothalamus, groups of GnRH neurons are found as early as 20 weeks of gestation, as well as in the adult human brain [9]. The hypothalamic population of GnRH neurons appears to continue rostrally in the TN in the adult and fetal human olfactory system. Standard immunocytochemical procedures seem to confirm the same origin from the olfactory placode. Through the cadaveric dissections of animals (South African clawed frogs, *X. laevis*) and subsequent immunocytochemical procedures, TN GnRH fibers were found in the olfactory bulb region. The fibers give rise to a dense plexus located ventrally to the bulb. Other fibers are also carried to more caudal levels of the telencephalon and diencephalon [10]. As another study shows, stimulating the peripheral trunk of the TN increases the levels of a GnRH-like compound in the cerebrospinal fluid of Atlantic stingrays [11]. This study was conducted in vivo on the species Dasyatis sabina by stimulating the peripheral nerve stem and analyzing the particles found in the cerebrospinal fluid [11]. In addition to GnRH, the olfactory pathway is also characterized by the presence of immunoreactive tyrosine hydroxylase. This cell population has been detected in the nasal region and the human embryonic telencephalon at the level of catecholaminergic neurons. These same areas are positive for GnRH research [12]. Considering GnRH, two forms of this molecule are present in the brains of all major vertebrate species. The study that demonstrated the presence of the two forms of GnRH was conducted on adult and juvenile lungfish (*Protopterus annectens*) using the high-performance liquid chromatography technique and radioimmunoassay with specific antisera [13]. As in the previous study, the analysis of GnRH led to the same conclusions, highlighting the presence of mammalian, salmon, and chicken II GnRH and various pituitary hormones, studied with double label immunocytochemistry [14]. Neuropeptide Y (NPY), instead, plays a key role in the regulation of gonadotropin-releasing hormone (GnRH), although its activity is not completely understood. A study of the forebrain of the teleost *Clarias batrachus* investigated one of the main roles of NPY in the regulation of GnRH. Dual immunocytochemistry shows associations and colocalizations of the two peptides in neurons of the olfactory system, and in the ganglion cells of the terminal nerve, as well as in the hypothalamus [15]. Finally, other molecules, such as Norepinephrine (NE), also have activity on TN. Norepinephrine (NE) has been shown to directly affect the TN activity level, and the ganglion response to electrical stimuli is influenced by both NE and acetylcholine (ACh), which demonstrated a variable effect on the total spectral power of TN activity. Given its activity, ACh can have both excitatory and inhibitory effects on TN ganglion cells [16].

### 3.4. Terminalis Nerve and Diseases

One of the pathognomonic symptoms of COVID-19 is olfactory dysfunction (Figure 3). The cause of this symptom is probably related to the reduced average volume of the olfactory bulb and tract in COVID-19 patients compared with the controls [17]. Theoretically, this can be explained by the neuroinvasive capacity of SARS-CoV-2. The hypothesis that SARS-CoV-2 travels via the olfactory route has not been confirmed. First, olfactory receptor neurons do not express the virus entry proteins ACE2 and TMPRSS2 and, therefore, are not infected, or extremely rarely infected, by SARS-CoV-2. This raises questions about the ability of SARS-CoV-2 to enter these neurons and travel along their axons into the brain. Moreover, the virus appears much more rapidly at downstream targets, which is inconsistent with axial transport and multiple transsynaptic transfers [18,19]. On the other hand, the nervus terminalis neurons express ACE2, and through the binding of the spike protein and this receptor, SARS-CoV-2 can infect these cells [17]. Furthermore, this cranial nerve serves as a direct connection between the olfactory epithelium and the hypothalamus, bypassing the olfactory bulb. Afterward, SARS-CoV-2 can penetrate the blood–brain barrier and can reach various neural circuits connected to the hypothalamus. This indicates that the nervus terminalis can be a route for the virus to reach the brain. This could explain why there is so much variability in the neuroinvasion of the brain, a characteristic that could not be explained by the classical route theory [18]. This hypothesis also justifies the rapid appearance of viral particles in the hypothalamus, which do not localize in the parenchyma of the olfactory bulb but are instead found at the superficial margin of the olfactory bulb, where the neurons of the terminal nerve reside [18,20,21]. The major study in favor of this theory is the one conducted by Bilinska et al. They studied nervus terminalis neurons in postnatal mice, double-labeled with antibodies against ACE2, gonadotropin-releasing hormone (GnRH), and choline acetyltransferase (CHAT), proving the route for the virus from the nasal epithelium, possibly via the innervation of Bowman’s glands, to brain targets, including the telencephalon and diencephalon [2]. In the end, biopsies of the olfactory epithelium from COVID-19 patients showed that the virus infected non-neuronal cells, defined as supporting cells of the olfactory epithelium. This implies that anosmia is caused by the loss of cell function support and not the loss of neurons in the olfactory bulb [18]. In the preceding section, we discussed the possible functions of the nervus terminalis, explaining its possible role in the reproductive system and how it modulates the olfactive system. The traumatic loss of the olfactory nerves is anecdotally associated with a reduction in libido [2]. Furthermore, it is thought that this nerve is implicated in Kallmann syndrome, a genetic form of hypogonadotropic hypogonadism. In more detail, in the embryonic phase, the failure of the migration of LHRH cells along the TN from the neural crest to the hypothalamus (through the olfactory placodes) results in the absence of LHRH cells, causing the diseases. Also, this syndrome is characterized by anosmia. The glycoprotein called “Anosmin-1”, encoded by the KAL1 gene, is defective in human Kallmann syndrome [4]. The neuromodulatory role of the terminal nerve in reproductive behavior via GnRH establishes an important link with hypothalamic nuclei, specifically the preoptic (POA) and the infundibular (INF) nuclei, which form the “kisspeptin neuronal (KP) network” [6]. Over time, the KP circuit has been associated with various biological functions. Through the release of GnRH from the hypothalamus, it is linked to diverse endocrinological conditions such as sexual development and human reproductive functions. The KP network regulates the response of gonadotropins (FSH and LH), inducing the synthesis and release of hormones crucial for reproduction. There are suggestions that the TN may trigger hormonal responses independently of other pathways, such as the KP neural circuit. In addition to this, the kisspeptin system participates in various circuits within the limbic system that mediate anxiety, fear, other negative emotions, and olfaction [1]. The afferent neurons of the KP network are poorly studied. The TN has many projections like the nasal mucosa and the amygdala, but also the hypothalamus. Projections that reach one or both hypothalamic nuclei may represent a potential afferent component to the KP neurons regulating GnRH secretions. To date, this interesting hypothesis is only speculation [6].

## 4. Discussion

The TN is well-developed in vertebrates, but it exists only as a ‘residue’ in the adult human brain (Figure 3). This evolutionary divergence may reflect a different “olfactory ecosystem,” driven by olfaction, climate change, and the potential evolution of the teleost fish forebrain [22]. Among the selected articles in this brief review, few deal with the human anatomy and physiology of the TN. However, some publications based on the study of the animal TN shows some interesting observations in humans, such as the migration from the olfactory placode to the central nervous system via the nasal septum. Also, studies conducted on embryos aim to demonstrate the presence of GnRH neurons in components of the adult and fetal human olfactory system [9]. According to this study, the migration of GnRH neurons in the fetal brain begins before the 10th week of gestation and these, once migrated, reach their maturity by the 20th week of gestation. To our knowledge, the projections, established during GnRH migration, continue to persist in the adult human olfactory system. The presence of GnRH neurons within the olfactory system in both the adult and fetal human brain, mirrors what has been found in other mammal species as well. Thus, GnRH neurons originate in the olfactory placode and migrate into the forebrain through the olfactory system. This also explains the hypogonadotropic hypogonadism that manifests as an olfactory deficiency in Kallmann syndrome [9]. Studies on Kallmann syndrome pathogenesis found that anosmin-1 (KAL1 gene) promotes axon outgrowth from the olfactory bulb and the branching into the olfactory cortex [23]. Furthermore, KAL1 is necessary for the contact between different structures (olfactory bulb and olfactory and vomeronasal axons, with regions of the telencephalon). These axons form the tracks that permit GnRH1 neurons to migrate into the brain. The loss of KAL1/Anosmin-1 could lead not only to olfactory dysfunction, but also to hypogonadotropic hypogonadism, which is observed in Kallmann syndrome [23].

Further studies showed the role of the TN. The pioneer neurons composing the TN showed expression of Prokineticin 2/Prokineticin-Receptor-2 (Prokr2) genes, loss-of-function mutations of which are responsible for this syndrome [24]. These data underline that a correct genetic expression of the TN Pioneer neurons guarantees the correct development of the olfactory system. Studies carried out on mouse models showed abundant gene expression in the single-cell RNA of mutated Prokr2 genes in animals affected by Kallmann syndrome [24]. Studying the mechanisms that correlate smell and the GnRH-1 system may open a new frontier for clinical research because little is yet known about the impact that the gonadal axis has on smell and vice-versa. Another consideration that we need to make is the importance of pheromones in humans. The fact that GnRH levels are altered by pheromones indicates that pheromones are not only important in mate selection, maternal behavior, and sexual arousal, but can also have a role in different pathologies such as infertility or sexual arousal difficulties. The role of pheromones in these pathologies has remained unexplored until now [7]. In vertebrates, the sense of smell plays a very important role in various behaviors including mate choice, food selection, homing, and escape from predators. The olfactory system is in close relationship with the limbic system, which is a region of the brain that plays an important role in various functions such as memory, emotions, interactions with the endocrine system, and learning. The interaction between neuromodulatory hormones and odor signals is at the basis of the behaviors that allow an animal to adapt to an environment [25]. These odor signals are also important for marine species, allowing them to adapt to the environment, as in the teleost. The anterior brain, also known as the Rhinoencephalon, receives many inputs from olfactory sources, which can integrate it and generate behavioral responses to stimuli within social, emotional, or motivational contexts crucial for survival, including mating, aggression, and defense. Like other sensory systems, the size of the neural tissue within the central nervous system is proportional to the importance of the function performed. Animals that rely heavily on olfactory signals have a markedly different proportion of dedicated space. In humans, 50% of the genes coded for olfactory receptors are present [25,26,27,28]. Returning to the terminal nerve, the extensive projections of TN GnRH neurons in the forebrain, together with their endogenous rhythmic activities, suggest that they may act in the global modulation of circuits to adapt to changes in hormonal or environmental conditions in the animal [29]. Moreover, a demonstration is that the TN GnRH3 network and other forebrain regions participate in the neuromodulation of the olfactory system and thus play an important role in other mammals and fishes [30]. LHRH is an important component of the TN in other mammals as well, with localization mainly in nasal areas and developmental stages [8]. It is known that circulating estrogenic levels regulate the activity of GnRH neurons; a decrease in GnRH has been observed following oophorectomy and, on the other hand, an increase in GnRH has been observed following the administration of estrogens, and similar observations have also been made in humans [10]. Since two forms of GnRH have been identified in animals, it could therefore be hypothesized that the forebrain and midbrain neurons could modulate species- and region-specific GnRH activity [13,14]. Since late 2019 to early 2020, many researchers have focused on COVID-19 with a wide view. The hypothesis that TN could be “the gate” for SARS-CoV-2 entering the brain was very interesting. Recent studies suggested that, as was previously stated, SARS-CoV-2 was isolated along the superficial portion of the olfactory bulb, where the terminal nerve is located, and not in the parenchyma (where the cells of the olfactory nerve are located) [18,21]. This could suggest that the terminal nerve may be a communication pathway with the central nervous system that has not yet been studied, with important physio-pathological meanings. As consequence, the TN could be used as a model to study viruses with a neuroinvasive capacity to fine-tune cognitive performance. The implication is that the nervus terminalis could also be an explanation for long COVID consequences, and post-COVID syndromes. Long COVID conditions could become an important public health issue in the future, considering that these post-COVID symptoms can last for years [31]. In this regard, neurological symptoms are very frequent and, in many cases, have an important impact on our patients’ lives. The most common symptoms are anosmia and cognitive decline, perceived as memory loss and difficulty concentrating [32,33]. Researchers have proposed different mechanisms of how SARS-CoV-2 could enter the brain and cause neuronal damage to explain the neurological symptoms of long COVID. A new study published by Sauve et al. [34] associates cognitive decline with low testosterone levels, due to alterations in the HPG (hypothalamic-pituitary-gonadal) axis, specifically GnRH secretion (reduction or alterations of the pulsating release of the hormone). The authors found that the association between anosmia, cognitive decline, and hypogonadism is like swhat they observed in Trisomy 21 (Down syndrome), which is known as a neurodegeneration process similar to Alzheimer’s disease [35]. The association between cognitive decline in Trisomy 21 and GnRH alterations was observed also in an animal model by Manfredi-Lozano and colleagues, where the replacement of GnRH improved cognition [36]. These observations plausibly suggest that the role of the TN might be larger than just as “a new route of entry into the CNS”. TN impairment could be the reason not only for anosmia but also for the HPG and GnRH alterations that determine hormonal changes that can be responsible for the cognitive deficits observed in long COVID patients. The additive aspect of anosmia and the neuroinvasive capacity of SARS-CoV-2 was elucidated by De Melo et al. [37]. According to these authors, infected animals, i.e., golden hamsters, had different symptoms depending on which virus they encountered. Regardless of the viral variants, all forms are neuroinvasive. Thus, neuroinvasion and anosmia are independent phenomena of SARS-CoV-2 infection. (62.5% of animals infected with the main Wuhan strain presented anosmia while 12% were infected with Gamma and none (0%) of the animals infected with Delta and Omicron/BA.1 showed signs of anosmia). Also, another factor is the viral load with which the hamster is inoculated. The study shows that despite the presence of the virus in the olfactory bulbs, animals infected with a lower infective dose had a lower incidence of olfactory changes [37]. Infection of the olfactory bulb is common regardless of the variant, but olfactory dysfunction is not as common. An inflammatory response was observed in the olfactory bulb at the same time (regardless of variant), but even the inflammation of this region is not sufficient to explain why in some forms, odor perception remains the same in the golden hamster. One idea may concern the dysregulation of the olfactory mucosa, as recovery is related to the regeneration of the olfactory epithelium in hamsters [37]. Could the TN also influence the variability of the olfactory dysfunction? It is very important to note that the timing of the deciliation and odorant receptor gene downregulation is fundamental to untangling the route of COVID-19 infection [38].

## 5. Conclusions

Most of the scientific studies on the terminal nerve are conducted on vertebrates. In recent years, there has been an increased interest in identifying the anatomy and physiology of the TN in humans, as it is a highly conserved neural structure. Comparing studies conducted on animals, the terminal nerve is not a vestigial structure as was previously believed, but has a well-defined role, even in humans. The plexiform organization of the TN could have a significant role in the development of the GnRH system, in the modulation of smell, and the physiology of the reproductive system. The more knowledge we have about the terminal nerve, the more we will be able to understand human reproduction, the olfactory system, and their diseases. The need for new anatomical knowledge is important and that the exact location of the TN in humans can be very important to avoid complications in ENT surgeries [39]. In animals, lesions of the nerve were associated with GnRH deficiency [6]. ENT surgeons are essential to the advancement of this knowledge. In conclusion, a better understanding of nerve anatomy and an easier way to identify this structure in vivo are needed.

## Figures and Tables

**Figure 1 ijms-25-03920-f001:**
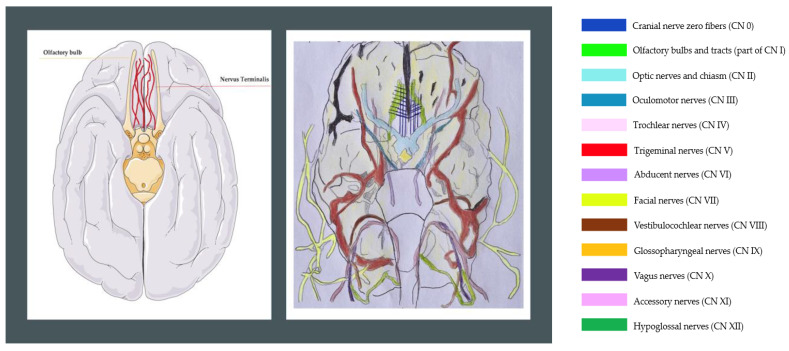
Representation of the ventral aspect of the human brain. The terminal nerve is reticulated in red (digital image on the left). On the right, the terminal nerve (TN) is reticulated in blue. The bilateral plexus of nerve fascicles of nervus terminalis covering the gyrus rectus of the human brain. This is an original drawing, showing the TN, during a surgery performed as part of the Specialization Program in Otolaryngology, La Sapienza University, Rome. “The accuracy of the handwriting cannot be assured since there is no verification”.

**Figure 2 ijms-25-03920-f002:**
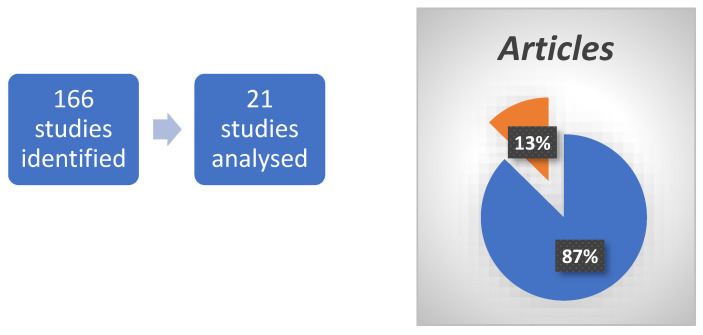
Left: From 166 studies 145 were excluded as non-relevant for our study; these include those in non-English languages and those with full text not available. Right: blue represents excluded articles; orange represents included and analyzed articles, as a percentage.

**Figure 3 ijms-25-03920-f003:**
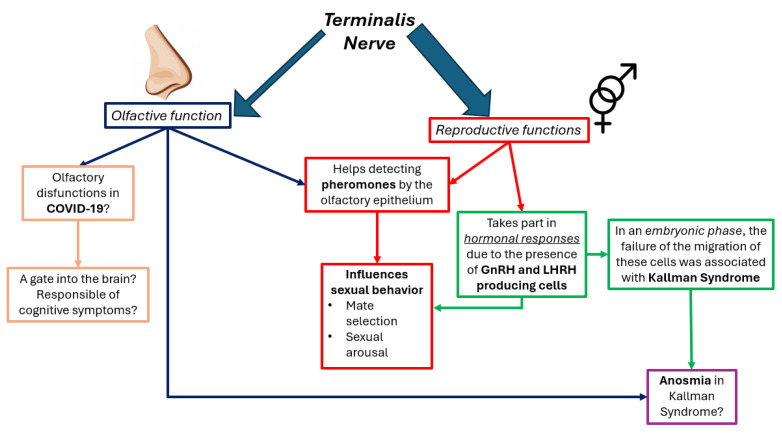
The figure illustrates the new proposed functions and pathological correlations of the terminalis nerve (TN). The first arrows have a different shape, demonstrating that most of the studies propose a more impactful role of TN in the reproductive function than in the olfactive one. Nevertheless, one does not exclude the other. Also, in this scheme associations between these two functions are present. Different colors were used: the left boxes (salmon color) show the hypothesis that COVID-19 is associated with the TN, while the red boxes show the role of this nerve in reproduction. The green boxes explain a possible etiology of Kallmann syndrome, in which a characteristic symptom is anosmia (violet box).

**Table 1 ijms-25-03920-t001:** This table summarizes the results extracted from the selected 21 articles. A summary of each is reported.

REFERENCE	ABSTRACT
[1]	The TN appears to have the same origin as the olfactory cells: the neural crest. Like other cranial nerves, its embryonic origins appear to lie in synergistic interactions during development between the neural and sensory crest placode.
[2]	Through the study of TN neurons in postnatal mice infected with a virus that traverses the olfactory pathways, an increase in gonadotropin-releasing hormone (GnRH) and choline acetyltransferase (CHAT) was identified.
[4]	The TN begins to develop at the edge of migrating neural crest cells with the olfactory and adenohypophyseal placodes.
[6]	The neural crest contributes to the subset of neurons that secrete GnRH. The TN neurons, indeed, appear to originate from the neural crest.
[7]	The TN seems to play a role in the olfactory function and in the reproductive function through the secretion of LHRH. Indeed, in women, the sense of smell is most acute during ovulation.
[8]	The TN shows a similar distribution of LHRH in both juvenile and adult animals. However, most of LHRH activity is greater in the adult brain.
[9]	Many studies on the fetal nervous system of animals have demonstrated the presence of cells that release gonadotropins, such as gonadotropin-releasing hormone (GnRH). The presence of gonadotropic cells present on the TN fibers was analyzed.
[10]	Through cadaveric dissections of animals and subsequent immunocytochemical procedures, TN GnRH fibers were found in the olfactory bulb region.
[11]	In studies conducted on Atlantic stingrays, by stimulating the peripheral nervous trunk and analyzing the particles present in the cerebrospinal fluid, the levels of a compound like GnRH increase in the TN.
[12]	GnRH, the olfactory pathway is further distinguished by the existence of immunoreactive tyrosine hydroxylase. This cellular population has been observed within the nasal region and the human embryonic telencephalon, specifically among catecholaminergic neurons. These identical regions display positivity in GnRH investigations.
[13]	In relation to GnRH, both forms of this molecule are found in the brains of all significant vertebrate species. The research that confirmed the existence of these two forms of GnRH was carried out on adult and juvenile lungfish (Protopterus annectens) utilizing high-performance liquid chromatography and radioimmunoassay with specialized antisera. Given the identification of two forms of GnRH in animals, it could be hypothesized that forebrain and midbrain neurons might regulate species- and region-specific GnRH activity [13,14]
[14]	Analysis of GnRH highlighting the presence of mammalian, salmon, and chicken II GnRH and various pituitary hormones. From this analysis, both sGnRH and mGnRH appear.
[15]	Neuropeptide Y (NPY) plays a key role in the regulation of gonadotropin-releasing hormone (GnRH).The study shows associations and colocalizations of GnRHs in the ganglion cells of the terminal nerve, as well as in the hypothalamus.
[16]	Norepinephrine (NE) also exhibits activity on the TN activity level, and ganglion response to electrical stimuli is influenced by both NE and acetylcholine (ACh). ACh can have both excitatory and inhibitory effects on TN ganglion cells.
[17]	It was hypothesized that olfactory dysfunction in patients with COVID-19 may be correlated with the reduced average volume of the olfactory bulb and tract. The neurons of the terminal nerve express ACE2, and through the binding of the spike protein and this receptor, SARS-CoV-2 can infect these cells.
[18]	TN serves as a direct connection between the olfactory epithelium and the hypothalamus, bypassing the olfactory bulb. This indicates that the nervus terminalis can be a route for SARS-CoV-2 to reach the brain. This could explain why there is so much variability in the neuroinvasion of the brain, a characteristic that could not be explained by the classical route theory. The hypothesis that SARS-CoV-2 travels through the olfactory pathway has not been confirmed. This is because olfactory receptor neurons do not express ACE2 and TMPRSS2, which are the proteins which the virus penetrates. Therefore, they are not infected, or very rarely. For this reason, there are doubts about the ability of SARS-CoV-2 to use this pathway [18,19]. In the end, biopsies of the olfactory epithelium from COVID-19 patients showed that the virus infected non-neuronal cells. This has made it clear that anosmia is caused by the loss of cell function support and not the loss of neurons in the olfactory bulb [18]. This could justify the rapid appearance of viral particles in the hypothalamus, which do not localize in the parenchyma of the olfactory bulb but are instead found at the superficial margin of the olfactory bulb, where the neurons of the terminal nerve reside [18,19,20].
[19]	As reported in [18]
[20]	As reported in [18]
[21]	As reported in [18]

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
