# Peer review of "Smelling TNT: Trends of the Terminal Nerve"

_ijms, 2024, doi:10.3390/ijms25073920_

Round 1

Reviewer 1 Report

Comments and Suggestions for Authors

I appreciate the opportunity to review the manuscript for publication in MDPI IJMS. I feel that the topics are interesting as for anatomy, functional hypotheses, and the possible pathological implications of the terminal nerve related in human diseases. However, the manuscript remains premature which should be thoroughly modified.

I have a few comments as follows.

In abstract, “Others suggest a more important role in reproduction and sexual behaviors. New emerging information suggests a possible role in pathologies like Kallman Syndrome, and COVID-19.”

A set of illustrations are necessary to clearly review these important points.

Figure 1: The authors describe that “This is an original drawing, which shows cranial nerve 0 (CN0), as nervous terminalis (NT), performed during exercise surgery of the Medical Residency Program in Otorhinolaryngology, Sapienza University, Rome.”

However, the accuracy of the hand writings cannot be assured since there is no verification.

L89: In the end, 145 studies were excluded, while the remaining 21 were analyzed and discussed in this review.

The authors should create a Table to summarize the results extracted from the selected articles.

In section of “3.2. Neurophysiology and functional aspects”

The author did not specify the Ref 7, which is a key article in the section.

In 3.3. Neuronal Immunochemical Studies of the Nervus Terminalis.

All references cited in the section are published in previous centuries decades before.

L176: “On the other hand, the nervus terminalis neurons express ACE2, and through the binding of the spike protein and this receptor, SARS-CoV-2 can infect these cells [17].”

Ref 17 is a review rather than an original article. It is inadequate to cite in the literature review.

The discussion part from L281 is apart from the topics of the Nervus terminalis.

Author Response

Reply to the Reviewer # 1 comments:

Dear Editor,

We thank the reviewer for his/her letter and the comments on our manuscript (Manuscript ID: ijms-2939941). Those comments are all valuable and very helpful for revising and improving our paper, as well as the important guiding significance to our research. Revised portions are marked in red in the manuscript. For the sake of clarity, requested point-by-point responses are included in the word manuscript file.  The main corrections in the manuscript and the response to the reviewer's comments are as follows:

REV1 I appreciate the opportunity to review the manuscript for publication in MDPI IJMS. I feel that the topics are interesting as for anatomy, functional hypotheses, and the possible pathological implications of the terminal nerve related in human diseases. However, the manuscript remains premature which should be thoroughly modified.

I have a few comments as follows.

In abstract, “Others suggest a more important role in reproduction and sexual behaviors. New emerging information suggests a possible role in pathologies like Kallman Syndrome, and COVID-19.”

A set of illustrations are necessary to clearly review these important points.

Reply1: We appreciated your suggestions and a new figure, Fig.2, which summarizes the new emerging trends in the Terminal Nerve, was added to the manuscript

1)    Figure 1: The authors describe that “This is an original drawing, which shows cranial nerve 0 (CN0), as nervous terminalis (NT), performed during exercise surgery of the Medical Residency Program in Otorhinolaryngology, Sapienza University, Rome.”

However, the accuracy of the hand writings cannot be assured since there is no verification.

 R1.1 R1: We completed the Fig.1 description adding the sentence “the accuracy of the handwritings cannot be assured since there is no verification”.

2)    L89: In the end, 145 studies were excluded, while the remaining 21 were analyzed and discussed in this review.

The authors should create a Table to summarize the results extracted from the selected articles.

 R1.2 R1: the Table 1, a Table summarizing the results extracted from the articles, was added.

3) In section of “3.2. Neurophysiology and functional aspects”

The author did not specify the Ref 7, which is a key article in the section.

R1.3 R1. The full citation was replaced: “7. Wirsig-Wiechmann CR. Function of gonadotropin-releasing hormone in olfaction. Keio J Med. 2001 Jun;50(2):81-5. doi: 10.2302/kjm.50.81.”

4)    In 3.3. Neuronal Immunochemical Studies of the Nervus Terminalis.

All references cited in the section are published in previous centuries decades before.

R1.4 All citations in this section refer to early immunochemical work on TN. We cited these references because it is important to show the reader the real evolution of the literature in this field, which is very time-limited.

5)    L176: “On the other hand, the nervus terminalis neurons express ACE2, and through the binding of the spike protein and this receptor, SARS-CoV-2 can infect these cells [17].”

Ref 17 is a review rather than an original article. It is inadequate to cite in the literature review.

R1.5 As suggested by the reviewer, the reference (Ref. 17), was replaced by: Bilinska K, von Bartheld CS, Butowt R. Expression of the ACE2 Virus Entry Protein in the Nervus Terminalis Reveals the Potential for an Alternative Route to Brain Infection in COVID-19. Front Cell Neurosci. 2021 Jul 5;15:674123. doi: 10.3389/fncel.2021.674123. 

-The discussion part from L281 is apart from the topics of the Nervus terminalis.

R1.6 As suggested by the reviewer, the sentences from 285 to 294 have been eliminated, to focus the discussion as reported in the manuscript.

Thanks again for the reviewer's comments. We revised each quoted sentence again.

We sincerely appreciate the warm work of the Reviewer and hope that our editing of this article can gain your valuable recognition, which is of great importance to us.

Kind regards,

Christian Barbato

Christian Barbato M.D. PhD,

National Research Council (CNR)

Institute of Biochemistry and Cell Biology (IBBC)

University Sapienza of Rome,

Viale del Policlinico, 155

00161 Rome, Italy

E-mail: christian.barbato@cnr.it

Reviewer 2 Report

Comments and Suggestions for Authors

Dear Authors,

 First, I must admit to being drawn to the proposed text about the terminal nerve. The idea of CN 0, or the elusive pheromone-related nerve seen in animals, has been more or less popular. The real-life evidence consists of animal studies, which sometimes are quick and sometimes difficult to relate to humans.

It is precisely where the main general remark is. As a reviewer, I respect the amount of work you have invested in the review. However, it is challenging to attain absolute credibility if the conclusion drawn is sometimes based on presumptions that are not easily proven. I would propose some changes in the Discussion and the Conclusion. Further remarks are as follows:

11. Line 2: I understand the concept of olfaction and chemical sensing, including the proposed role of CN 0, but I do not understand the relationship with TNT. Maybe Smelling trends of the terminal nerve or similar.

2.      Line 39: You have stated the term nervus terminalis (NT), which is fine, but later you also use TN and terminalis nerve. Please use the same term and abbreviation.

3.      Line 68-69: The sentence refers to the study of the Roussel et al. However, the sentence lacks factual information. The following sentence should be fused to the previous one to show better the information Roussel et al. have confirmed in their work on the landmarks of the endonasal skull base surgery.

4.      Line 75: See remark 2.

5.      Line 164-182: The involvement of the SARS-CoV-2 in the olfactory epithelium and the questionable first reports in the COVID-19 era stating different target cells, including neuroinvasive capacity, is known. The authors have stated all the relevant proposals, including the possible route of viral spread to the brain. I would mildly propose simplifying the text, focusing more on the mucosal and neuroinvasive scenarios with all the same references, not confusing the reader.

6.      Lines 192-193: See remark 5. The information regarding the mucosal and neuroinvasive scenarios is fragmented throughout the text.

7.      Line 216: See remark 2.

8.      Line 220: Please change Figure 1. It does not allow the reader to see all the intended information in its present form.

9.      Line 252: See remark 2.

10.  Line 308-312: Please rewrite the segment. Please start with the problem and end with the solution, not vice versa.

11.  Line 319-322: Please rewrite. I do understand the intention, but the readers may struggle.

12.  Line 325: See remark 2.

13.  Line 350: See remark 2.

14.  Line 359: I could not refer the statement to any referenced text in the review. Please add the needed text to the review.

15.  Line 360: The statement is self-explanatory. It may not need to be in the conclusion.

16.  Line 390: Please use a proper reference

17.  References: Please check the use of small and all caps in this section.

Author Response

Reply to the Reviewer # 2 comments:

Dear Editor,

We thank the reviewer for his/her letter and the comments on our manuscript (Manuscript ID: ijms-2939941). Those comments are all valuable and very helpful for revising and improving our paper, as well as the important guiding significance to our research. Revised portions are marked in red in the manuscript. For the sake of clarity, requested point-by-point responses are included in the word manuscript file.  The main corrections in the manuscript and the response to the reviewer's comments are as follows:

REV1 Dear Authors,

 First, I must admit to being drawn to the proposed text about the terminal nerve. The idea of CN 0, or the elusive pheromone-related nerve seen in animals, has been more or less popular. The real-life evidence consists of animal studies, which sometimes are quick and sometimes difficult to relate to humans.

It is precisely where the main general remark is. As a reviewer, I respect the amount of work you have invested in the review. However, it is challenging to attain absolute credibility if the conclusion drawn is sometimes based on presumptions that are not easily proven. I would propose some changes in the Discussion and the Conclusion. Further remarks are as follows:

  1. Line 2: I understand the concept of olfaction and chemical sensing, including the proposed role of CN 0, but I do not understand the relationship with TNT. Maybe Smelling trends of the terminal nerve or similar.

R2.1 TITLE: We thank the reviewer for the helpful suggestion, but the idea to include the acronym TNT, Trends of the Terminal Nerve, refers to the acronym for the compound, TNT Trinitrotoluene, a well-known explosive that is often identified through the use of dogs' extraordinary sniffing abilities. Furthermore, given that the study of smell has largely returned to the foreground after the pandemic, the goal was to create curiosity about the terminal nerve by imagining that more intense scientific research on the topic could be 'explosive' for the innovations it could bring. We hope that the explanation was clear and comprehensive.

  1. Line 39: You have stated the term nervus terminalis (NT), which is fine, but later you also use TN and terminalis nerve. Please use the same term and abbreviation.

R2.2 Line 39: The sentence was replaced by “The Latin name of the nervus terminalis has now been currently replaced by terminal nerve and terminalis nerve (TN) indicating a rudimentary structure found in humans and higher mammals, which can be found in fetal stages.”  The same term and abbreviation (TN) were used in the manuscript.

  1. Line 68-69: The sentence refers to the study of the Roussel et al. However, the sentence lacks factual information. The following sentence should be fused to the previous one to show better the information Roussel et al. have confirmed in their work on the landmarks of the endonasal skull base surgery.

R2.3 Line 68-69: as suggested by reviewer, the sentence was replaced by “Roussel et al. have confirmed in their work on the landmarks of the endonasal skull base surgery.”

4 Line 75: See remark 2.

R2.4 The term was replaced with TN.

  1. Line 164-182: The involvement of the SARS-CoV-2 in the olfactory epithelium and the questionable first reports in the COVID-19 era stating different target cells, including neuroinvasive capacity, is known. The authors have stated all the relevant proposals, including the possible route of viral spread to the brain. I would mildly propose simplifying the text, focusing more on the mucosal and neuroinvasive scenarios with all the same references, not confusing the reader.

R2.5 The full paragraph was rewritten and several sentences were eliminated and replaced with:

“Probably the cause of this symptom is related to the reduced average volume of the olfactory bulb and tract in COVID-19 patients compared with the controls [17], theoretically by the neuroinvasive capacity of SARS-CoV-2. The hypothesis that SARS-CoV-2 travels via the olfactory route has not been confirmed. First, olfactory receptor neurons do not express the virus entry proteins ACE2 and TMPRSS2 and, therefore, are not infected, or extremely rarely infected, by SARS-CoV-2. This raises questions about the ability of SARS-CoV-2 to enter these neurons and travel along their axons into the brain. Another reason is that the virus appears much more rapidly at downstream targets than is consistent with axial transport and multiple transsynaptic transfers [18,19].”

  1. Lines 192-193: See remark 5. The information regarding the mucosal and neuroinvasive scenarios is fragmented throughout the text.

R2.6. See point R2.5

  1. Line 216: See remark 2.

R2.7 The term was replaced with TN.

  1. Line 220: Please change Figure 1. It does not allow the reader to see all the intended information in its present form.

R2.8 We are aware of and accept the criticisms of Figure 1. Conversely, the activity of future otolaryngologists and neurosurgeons may also include the need to draw, without any technological aid, anatomical images in university exercises. Even if made in a non-exemplary form, as abundantly present in the literature and created with technological systems, this image is very close to a 'real view of the anatomical district', and not created like an anatomy manual. Exceptional anatomists such as Frank H. Netter, an American doctor and illustrator, and author of well-known atlases of human anatomy, can be counted on the fingers of one hand.

  1. Line 252: See remark 2.

R2.9 The term was replaced with TN.

  1. Line 308-312: Please rewrite the segment. Please start with the problem and end with the solution, not vice versa.

R2.10 The sentence was rewritten as suggested: “Researchers have proposed different mechanisms of how SARS-CoV-2 could enter the brain and therefore cause neuronal damage, to explain the neurological symptoms of Long COVID.”

  1. Line 319-322: Please rewrite. I do understand the intention, but the readers may struggle.

R2.11 The sentence was eliminated and replaced with: These observations may plausibly suggest that the role of the TN might be different from that of being just “a new route of entry into the CNS”.

  1. Line 325: See remark 2.

R2.12 The term was replaced with TN.

  1. Line 350: See remark 2.

R2.13 The term was replaced with TN.

  1. Line 359: I could not refer the statement to any referenced text in the review. Please add the needed text to the review.

R2.14 A new reference was added to the text: Peña-Melián Á, Cabello-de la Rosa JP, Gallardo-Alcañiz MJ, Vaamonde-Gamo J, Relea-Calatayud F, González-López L, Villanueva-Anguita P, Flores-Cuadrado A, Saiz-Sánchez D, Martínez-Marcos A. Cranial Pair 0: The Nervus Terminalis. Anat Rec (Hoboken). 2019 Mar;302(3):394-404. doi: 10.1002/ar.23826.

  1. Line 360: The statement is self-explanatory. It may not need to be in the conclusion.

R2.15 The sentence was eliminated.

  1. Line 390: Please use a proper reference

R2.64 An appropriate reference was used.

Thanks again for the reviewer's comments. We revised each quoted sentence again.

We sincerely appreciate the warm work of the Reviewer and hope that our editing of this article can gain your valuable recognition, which is of great importance to us.

Kind regards,

Christian Barbato

Christian Barbato M.D. PhD,

National Research Council (CNR)

Institute of Biochemistry and Cell Biology (IBBC)

University Sapienza of Rome,

Viale del Policlinico, 155

00161 Rome, Italy

E-mail: christian.barbato@cnr.it

Round 2

Reviewer 1 Report

Comments and Suggestions for Authors

I appreciate the opportunity to review again the manuscript for publication in MDPI IJMS. I reckon that the manuscript has been revised and improved in accordance with the reviewers’ comments.

The authors should create a Table to summarize the results extracted from the selected 21 articles. I say again, a brief sammary for each is necessary. 

Author Response

Reviewer 1 II ROUND

I appreciate the opportunity to review again the manuscript for publication in MDPI IJMS. I reckon that the manuscript has been revised and improved in accordance with the reviewers’ comments.

The authors should create a Table to summarize the results extracted from the selected 21 articles. I say again, a brief sammary for each is necessary. 

Reply Reviewer 1 II ROUND

Thanks again for the reviewer's comments. We summarize the results extracted from the 21 articles in Table 1.

We sincerely appreciate the warm work of the Reviewer and hope that our editing of this article can gain your valuable recognition, which is of great importance to us.

Reviewer 2 Report

Comments and Suggestions for Authors

Dear Authors, 

You have successfully answered all of the remarks. 

Author Response

Reviewer 1 II ROUND

Dear Authors, 

You have successfully answered all of the remarks. 

Reply Reviewer 1 II ROUND

We sincerely appreciate the warm work of the Reviewer and hope that our editing of this article can gain your valuable recognition, which is of great importance to us.